# Plastic Mulch Films in Agriculture: Their Use, Environmental Problems, Recycling and Alternatives

Kotaiba Salama [1] and Martin Geyer [2,*]

1.  Leibniz Centre for Agricultural Landscape Research, 15374 Müncheberg, Germany; kotaiba.salama@zalf.de
2.  Leibniz Institute for Agricultural Engineering and Bioeconomy, 14469 Potsdam, Germany
*   Correspondence: mgeyer@atb-potsdam.de

**Abstract:** Agricultural plastic mulching is an important horticultural process for increasing crop yields because it preserves soil moisture, soil temperature, and nutrients, and avoids the need for weed herbicides. However, there are risks to using plastic mulch, since residual macroplastic (MaP), microplastic (MP), and nanoplastic (NP) in fields have a significant negative impact on the environment, causing damage to soil properties, harming microorganisms in the soil, and entering the human body via the food chain. Plastic mulch is often disposed of in landfills or used in techniques like the thermal process to gain energy or recycling to generate plastic granules for the plastic industry. Pretreatments are occasionally required before recycling, such as cleaning the mulch from the soil to fit the recycling process. This review provides an overview of the quantities and negative impacts of plastic, especially plastic mulch films after use, as well as their decomposition products, on the environment, soil, and human health, and presents alternatives. The possibilities and problems of collecting and recycling films are discussed in addition to the alternatives, for example, the use of biodegradable films. Overall, agricultural advancements to reduce plastic waste in the environment by using thicker films, collection after use, and recycling in developed countries are on a good path. However, NP poses a risk, as it is still completely unclear how it affects human health. Alternatives to plastic mulch have found little acceptance so far due to the significantly higher material costs.

**Keywords:** horticulture; mulch; plastic; microplastic; environment; soil; pollution; disposal; recycling; alternatives

## 1. Introduction

Mulching is a crucial horticultural technique to increase crop productivity [1]. Plastic's success is due to lower cost, lighter weight, ease of use, minimal installation and management costs, and great durability [1]. In addition, mulch raises the soil temperature, which changes the physical and chemical composition of the soil, promoting early harvest and faster crop development. Moreover, it improves plant growth speed and increases water retention capacity [2–6]. Additionally, mulch protects the soil's surface against unfavorable causes like erosion and improves the conditions for vegetables to grow by preventing surface runoff of water and reducing nitrogen and fertilizer leaching [7–12]. Therefore, mulch is often used in fruit and vegetable cropping systems around the world because of the aforementioned benefits [6,13–16]. Moreover, opaque mulching helps prevent the growth of weeds and thus reduces the use of chemical pesticides that cause environmental pollution [17,18]. Acharya et al. [19] and Muhammad et al. [20] observed that mulch treatments significantly boost the total nitrogen, phosphate, and potassium uptake compared to identical non-mulched plants.

The microclimate around vegetable plants is influenced by the color of the mulch and how much light it transmits [21,22]. This influences the soil temperature underneath the film, where black plastic mulch increases soil temperature by between 4 and 6 degrees Celsius at depths of 5 cm and 10 cm, respectively [2]. Numerous studies have revealed that

using plastic mulch greatly enhances crop productivity. Gao et al. [23] examined the effects of plastic mulching on the production of cotton, wheat, potatoes, and maize in China and found that it greatly increased yields (24.3% on average) and improved water consumption efficiency (27.6% on average). The use of plastic mulch can increase grain crop output by 20% to 35% [24]. Raising plant density from 65,000 to 85,000 plants/ha using mulch can increase grain yield by 0.6 to 1.2 t/ha [25]. Accordingly, Shiukhy et al. [15] showed that mulch application led to the highest marketable strawberry yield when compared to the control. The average marketable yield increased by 25–28% in transparent plastic mulch and 15% in black plastic mulch when compared to the control. Low-density polyethylene (LDPE) film, which is primarily non-biodegradable, is used in commercial crop production. It is available in a variety of colors, transparent or opaque, and has a relatively low cost [26,27]. The most used thickness of low-density polyethylene (LDPE) films is 31 μm in the USA. The Chinese government is encouraging and financing the manufacturing and use of thicker films with a thickness of 10 μm, compared with previously used films with a thickness of only 6 μm, which has increased the recovery ratio from 70% to 90% [28,29]. In Europe, mulch film has a thickness usually between 15 and 80 μm covering the soil in crop production systems annually or perennially [30,31]. In China, policies for the management of agricultural films, established by the Chinese government in 2020, are intended to encourage the use of biodegradable mulch and thicker plastic mulch films (>10 μm), which are required under these policies [32]. Some crops, like white asparagus, need films with 100 μm and 150 μm thicknesses [30,33,34]. Mulch films have been utilized for approximately 8 years. Only for four months a year, during the harvest, do such films cover ridges [30,34]. The results from Xiong et al. [35] showed that, as the thickness of LDPE films increased, the proportion of the total area contaminated decreased dramatically. The highest and lowest damaged areas were 32.2% and 3.5% in 6 and 15 μm films, respectively.

In Europe, with a total plastic market of 57.9 Mt in 2019, 0.32 Mt is used for vegetable production, of which 0.083 Mt is used for mulch film (Figures 1 and 2) [36]. In Germany, Bertling et al. [37] showed that 1.1 Mt of plastic is consumed by agriculture each year (comprising around 560,000 tons of thermoplastics and duroplastics and approximately 540,000 tons of chemical fibers, polymer dispersion, and elastomers). It accounts for 4.7% of the total German consumption of 23.6 Mt plastic annually. Regarding plastic consumption, agriculture achieves a recyclate share of around 37%. It is thus well ahead of other plastic applications when implementing a circular economy.

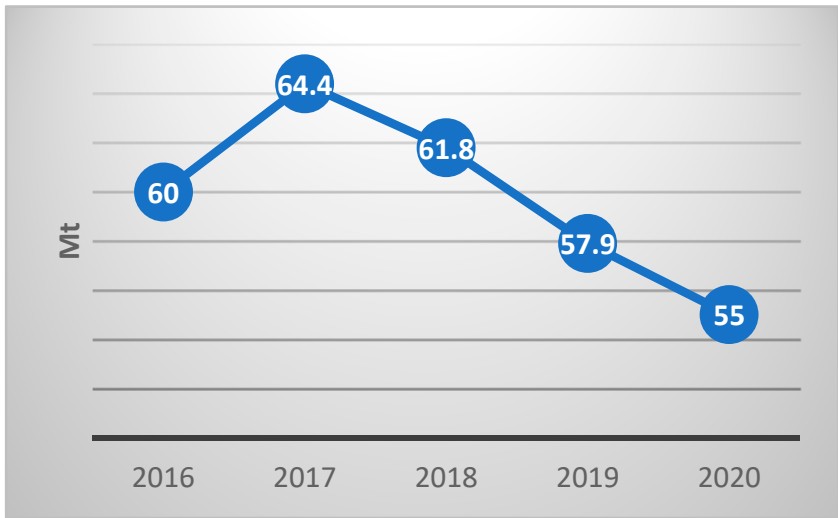

**Figure 1.** Plastic production (million tons) in EU, 2016–2020 [38].

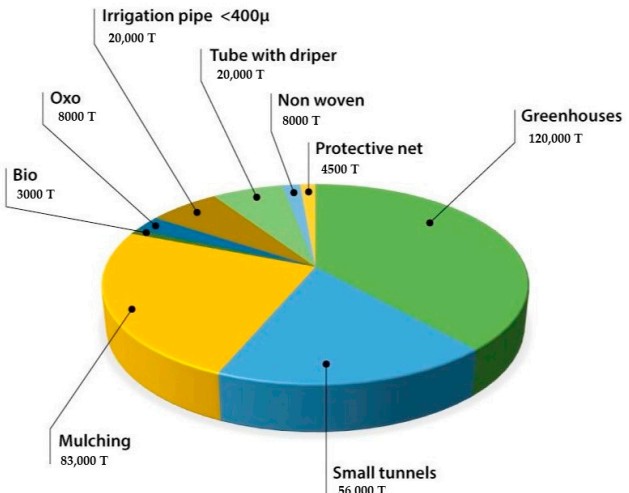

**Figure 2.** Plasticulture for vegetable production in the EU reached a total of 324,500 tons in 2019 [36] (Oxo = oxo-degradable plastics, Bio = biodegradable mulch).

## 2. Plastic in the Environment

In the ecology of terrestrial ecosystems, MP pollution in soil has been ranked as the second most significant scientific issue. Compared with other ecosystems, agricultural soil frequently has higher levels of MP pollution [39], and it can be up to 23 times worse than MP pollution in aquatic ecosystems [40]. PE mulch fragments in the soil degrade very slowly; depending on the soil and environment, full degradation may take 300 years [41,42]. Despite the advantages of using PE mulch in the production of fruits and vegetables, its use is not regarded as environmentally friendly because the conventional techniques through which farmers dispose of PE mulch frequently cause soil and air pollution [43].

According to the European Parliament [44], the share of landfills in the EU decreased from 24% in 2017 to 18% in 2020. In line with the EU Landfill Directive, EU countries must reduce the amount of municipal waste sent to landfills to 10% or less of the total municipal waste generated by 2035.

The disposal of waste and plastic waste in landfills, despite all the problems associated with it in terms of environmental pollution, is still a common practice worldwide. The practice of landfilling also continues in the eastern and southern parts of Europe. In Bulgaria and Malta, it is more than 70%. In Greece, Cyprus, and Romania, it is more than 50%, while it is now less than 50% in Spain and Portugal compared with 2017.

Between 2017 and 2020, landfilling decreased substantially in Croatia (31 percentage points), Poland (31 percentage points), Slovakia (30 percentage points), Cyprus, (30 percentage points), Greece (20 percentage points), Malta (20 percentage points), and Romania (20 percentage points) [44].

The use of plastic mulches in agricultural systems is fraught with several serious issues. For instance, concerns about polyethylene originate from its reliance on fossil fuels during manufacture, the ensuing waste problem, and the subsequent release of hazardous chemicals into the environment as a result of conventional disposal procedures. Polyethylene (PE) mulch films have a life span of typically one growing season before being discarded [35]. According to Zhang et al. [45], the average MP abundance at the research locations with varying mulching histories is 538, 1484, 5812, and 9708 particles/kg soil, respectively, for the period of mulching (5 years, 5–10 years, 10–20 years, and >20 years).

In addition, concerns about pesticide residues that may contaminate polyethylene mulch fractions that spread in fields have also been raised [46–48]. PE films may function as a pesticide vector, by absorbing the pesticide in the film's non-crystalline regions, which facilitates the movement of pesticides into the soil matrix [49–51]. Mulch that has been exposed to pesticides should not be disposed of in landfills, because there is a risk that the pesticides will leach out [52,53]. Ramos et al. [51] demonstrated a significant increase in

pesticide accumulation in the film compared with soil, with the accumulation ranging from 584 to 2284 μg of pesticide/g of plastic compared with 13 to 32 μg pesticide/g of soil. In addition, phthalate esters (PAEs) may build up in the soil as a result of the leftover plastic films [54–56]. Lü et al. [56] tested 50 agricultural films, and they were found to contain phthalate acid ester (PAE, plasticizer) in concentrations ranging from 2.59 to 282,000 mg/kg. According to Wang et al. [57,58], the average PAE concentration in non-mulched farmland was 0.37–0.73 mg/kg, whereas it was 0.45–0.81 mg/kg in the 0–25 cm soil layer of plastic mulched farmland.

Due to concerns about the disposal of contaminated plastic and the decreasing amount of land accessible for landfilling, the industrial incineration of plastics is a viable alternative. In OECD EU countries, energy recovery (incineration) is the most used method for plastic waste disposal, followed by landfill and recycling [44,59] (Figure 3). In many other countries, mismanagement and uncontrolled litter are still the most common issues associated with plastic waste disposal. According to the findings of Valavanidis et al. [60], all plastics burn easily and emit persistent radicals with a carbon and oxygen base that are known to have negative effects on inhalable airborne particles. In countries where landfills are poorly managed, or the alternative energy systems mostly depend on fossil fuels, incineration may be environmentally justified [61]. Both the residual solid ash and particle smoke emissions contain these radicals. Dioxins, furans, and other gaseous organic pollutants that can be harmful to human health are produced when plastics are burned [60,62–64]. According to Levitan and Barros [65], the illegal incineration of plastics on-site could result in levels of dioxins that are 40 times greater than ambient particulate matter and 20 times higher than controlled high-temperature incineration. Additionally, PE may be burned to create heat or electricity and has the same amount of potential energy per unit weight as oil [66,67].

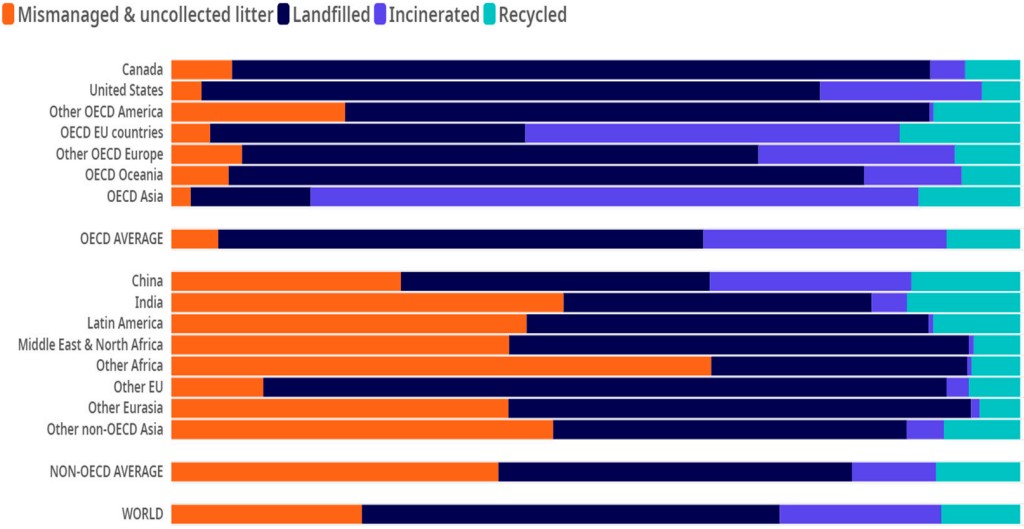

**Figure 3.** Share of plastics treated by waste management category, after disposal of recycling residues and collected litter [59]. The x-axis shows the different quantities in %.

## 3. Plastics in the Soil

Plastic waste is typically divided into three categories based on its size: nanoplastic (NP, <1 μm), microplastic (MP, 1–5 mm), and macroplastic (MaP > 5 mm) [68].

The term "nanoplastics" is still under debate, and different studies have set the upper size limit at either 1 μm or 0.1 μm [69]. According to Machado et al. [70], the types of polymers generated from MP particles that are dispersed in the soil differ according to their shape. For instance, the largest MP particles and source of fibers are polyesters (PESs). While PE pieces resemble the soil's natural soil particles in both size and shape, polyamide (PA) has the smallest size (15 μm) [70]. According to Bläsing and Amelung [71] and Tull et al. [72], the sources of MP in the soil can include industrial plastics, waste, road

dust, road wear particles, air deposition, and soil sedimentation [73,74]. Moreover, tire abrasion is one of the major sources of MPs in the environment. Although most tire particles settle into soils, studies on their ecotoxicological impacts on the terrestrial environment are scarce [75]. The concentration of MPs in the soil around an industrial site can be between 0.03% and 6.7%, according to Fuller and Gautam [76]. Concentrations of MPs up to 0.002% have been found in soils, even in practically human-free natural reserve areas [77]. These concentrations may be attributed to the nanoparticles resulting from the friction processes of plastic particles, such as the friction of car tires [78]. However, agriculture production is often seen in the public perception as the major cause of (micro)plastic pollution of soils [79]. Organic waste and compost, mulch films, wastewater irrigation, air deposition, and combined pesticides and fertilizers represent the majority of MP sources on agricultural lands [80]. One of the additional sources of MPs is the in situ degradation of big plastic fragments (MaPs). They increase the possibility of heterogeneous items, like plastic film or trash from the home, becoming the main source of plastic waste [81–83]. In terrestrial areas, plastic mulching is a substantial source of MPs because of its intensive use and improper cleanup [79,84,85]. If plastic mulches or parts of them are intentionally or unintentionally left in agricultural soils, they become brittle over time and finally disintegrate into MPs [84]. According to Huang et al. [86], who found a significant linear connection (R2 = 0.61) between the time of consumption and plastic residue in soils, the consumption of plastic mulch film may be a significant source of plastics. According to Bläsing and Ameling [71], Nizzetto et al. [87], and Steinmetz et al. [31], the poor management of mulch film disposal caused a significant amount of plastic mulch residues to persist in the soil. MPs accumulate as a result of the physical abrasion and UV-induced breakdown of plastic mulch. As previous studies have indicated that soils with a high level of MaPs may contain MPs as well because of the fragmentation or breakdown of MaPs as a potential source of MPs, it is imperative to monitor MaPs and MPs in agricultural regions [85,88,89]. Furthermore, Büks et al. [90] pointed out that the potential harmful effects of particles increase with the decrease in particle size. Agricultural soils may have an overload of secondary MPs as a result of frequent tillage procedures that include the incorporation of plastic film debris in soil mineral aggregates [31]. The high resistance of plastic particles to disintegration is predicted to result in the accumulation of 630 Mt of plastic rubbish and 120 Mt of MP waste on Earth by the year 2050 [91–93]. An important contributing reason to the high concentration of MPs in these soils is the application of extremely thin mulch films with thicknesses between 6 and 10 μm that are easily shredded. Zhang et al. [94] found that the areas mulched with the 6 μm film had a residual film mass greater than 75 kg/ha in >90% of the areas, while the percentage was only 48.8% for the 8 μm film mulching regions (Figure 4). He et al. [95] noted that MP contamination is commonly measured in kilograms per hectare. Li et al. [96] examined the MPs that had been collected after 32 years of the continuous application of plastic mulch film in an agricultural area at Shenyang Agricultural University in China. In the topsoil (0–10 cm), the total concentration of MP particles increased from 7183 to 10,586 particles/kg, with an average of 8885 particles/kg, while in the deep subsoil (80–100 cm), it increased from 2268 to 3529 particles/kg, with an average of 2899 particles/kg. Wang et al. [97] found that MP abundances in farmlands in five Chinese provinces ranged from 2783 to 6366 particles/kg in all samples, and MP distribution results showed that more than 80% of particles were less than 1 mm, with MP particles sizes between 0.02 and 0.2 mm constituting the largest proportion. MP abundances ranged from less than 80% to more than 80% in all samples. In a study on soils in 19 provinces throughout China, Huang et al. [86] found that higher MP particle concentrations were present in fields where plastic mulching had been used continuously for 24 years, with amounts of 31–129.6, 169–446.1, and 728–1422.4 particles/kg soil in fields with 5, 15, and 24 years of continuous mulching, respectively. MP contamination of 83.6 kg/ha, on average, was found in all of the soil samples collected from farmlands throughout China. Machado et al. [98] reported that farmland in southern China had between 7100 and 42,960 particles/kg MP. The majority of this MP is believed to originate from plastic

film mulching and sewage application, with 95% of the sampled plastic particles falling within the MP size range of 0.05–1 mm.

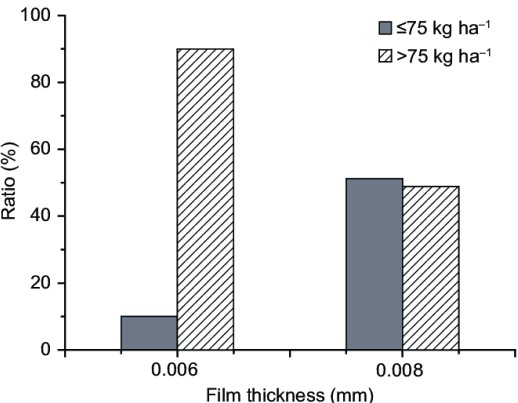

**Figure 4.** Effect of film thickness (e.g., 0.006 and 0.008 mm) on the amount of residual mulch film [94]. The y-axis shows the ratio of areas in % with > and <75 kg/ha MP.

### 3.1. Impact of MPs on Soil Physical Properties

According to Machado et al. [98] and Zhang and Liu [99], MP particles integrate into soil clumps and aggregate to varying degrees, affecting the physical properties of the soil. Machado et al. [70] investigated the effects of different MP morphologies (shapes) on soil properties (Figure 5). Polyester fibers (PES) increased water-holding capacity but PE and polyacrylic fibers (PMMA) did not follow a clear trend. Soil bulk density and soil microbial activity decreased for all three plastics, except for PE. Soil structure and function showed no clear trend for any of the three plastics. Lozano et al. [100] also observed that all MP forms decreased soil aggregation, which may be due to the possibility that MPs could generate fracture points in the aggregates, which would affect their stability, as well as due to potential negative impacts on soil biota. Boots et al. [101] indicated that MPs can change the way microaggregates connect and prevent the formation of large aggregates. According to Machado et al. [98], while PMMA microfiber with an average diameter of 18 μm lowered the quantity of soil water-stable aggregates, PES microfiber with an average diameter of 8 μm had the opposite effect. Furthermore, using meta-analysis, Lehmann et al. [102] found that the stability of soil aggregates and large aggregates was affected by the linear structure of PES microfibers.

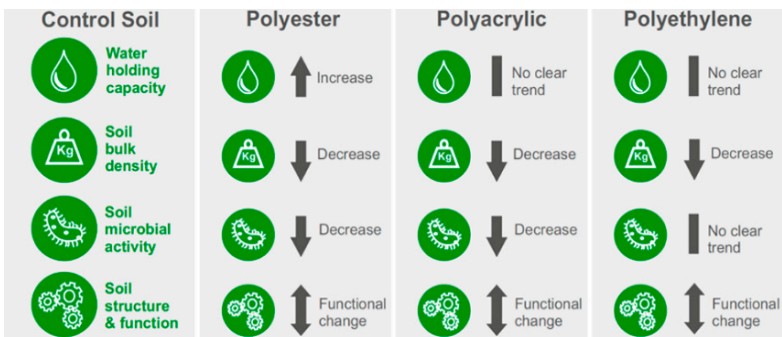

**Figure 5.** Effect of MP types (polyacrylic fibers, polyester fibers, and polyethylene fragments) on bulk density, water-holding capacity, hydraulic conductivity, soil aggregation, and microbial activity [98].

Other studies showed different results, possibly due to different soil types and different experimental conditions. Ingraffia et al. [103] found that MP fibers caused Vertisol's bulk density to decrease by 9%. Machado et al. [70,98] also reported that the bulk density of loamy sand soil decreased with the increase in polyester MP fiber concentration. By contrast, Koskei et al. [104] found that soil bulk density increased from 1.19 to 1.31 g/cm³

regardless of the type of residue, and the soil porosity decreased from 58% in the treatment without residual film to 57.4% in biodegradable film and 56.1% in LDPE significantly ($p < 0.05$). Zhang et al. [105] revealed no differences in soil bulk density between clay loam soil examined in field and greenhouse conditions. The contrast between soil particles, which predominantly consist of aluminosilicate minerals, and the lower density of MPs explained how MPs lowered soil bulk density, according to Machado et al. [70,98]. However, Zhang et al. [105] found no indication of a significant impact of PE microfibers on soil bulk density, most likely as a result of the low concentrations (0.3%). They discovered that when concentrations increased, the bulk density of polyester microfibers (PMFs) dropped.

The application of MPs to soils improves soil evaporation and evapotranspiration rates, demonstrating that MPs diminish soil water retention, according to several publications, e.g., [70,98,105–108]. As a result, soil moisture content decreases and irrigation utilizes more water. Significantly higher evaporation and evapotranspiration are related to particle size and concentration. Similar-sized particles in soil aggregates (i.e., 2 mm) can easily enter the soil profile and contribute to water migration channels [106,107], whereas medium particles (5–10 mm) can cover soil surfaces, limiting evaporation, at least initially [107]. However, larger MP particles (10–15 mm) can increase soil surface cracking, which can accelerate evaporation and soil desiccation [107]. According to a few studies on soil evaporation and desiccation, plastic may exacerbate soil ecosystems' tendency to dry out, particularly in desert areas. Additionally, as soil dries out, cracks and fissures form, which might promote the leakage of MP particles into groundwater and increase the MP pollution of aquatic organisms. Zhang et al. [105] discovered that, after one year, the 0.3% polyester microfiber (NPMF) treatment had a significantly higher volume of >30 μm pores (21.4%) than the 0.1% NPMF treatment (11.6%) and the control treatment (10.3%). Wan et al. [107] investigated the effect of the addition of plastic film fragments (2, 5, and 10 mm in size) at 0.5% and 1.0% concentration in terms of increasing water evaporation by producing water tunnels (Figure 6). As the particle size became smaller (1–2 mm), and the amount of plastic increased to 1%, there was a significant increase of more than 20% in evaporation. The remaining LDPE and biodegradable plastic film residues increased evapotranspiration and had a negative effect on water use effectiveness and soil porosity, as indicated by Koskei et al. [104]. In comparison, Rillig et al. [109] discovered that MPs can have a negative impact on the water cycle in soils, worsen soil water shortages, and affect the transport of pollutants into deep soil layers.

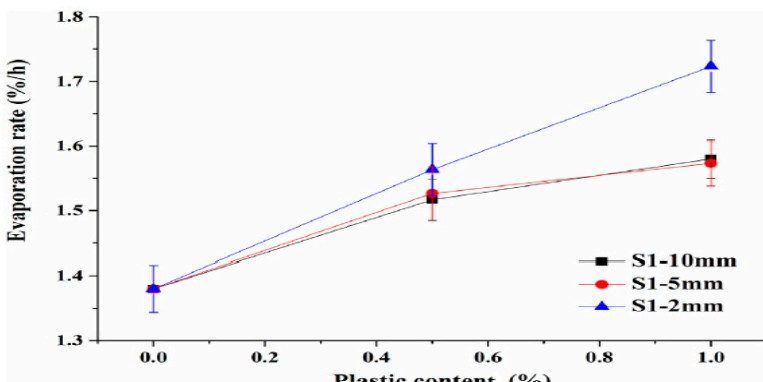

**Figure 6.** Changes in the evaporation rate in soil treated with different amendment ratios of plastics (S1 = size of plastic particles) [107].

### 3.2. Impact of MPs on Soil Fertility, Soil Carbon, and Plant Growth

MPs can significantly affect soil fertility through several factors; with higher soil MP content, fertility is reduced or soil fertility measurements are affected. High catalytic capacity soil enzymes play a key role in regulating the cycling of basic nutrients in the soil and are closely associated with some soil biochemical activities [110]. According to the studies by Liu et al. [111] and Huang et al. [112], polypropylene-derived MPs

have a significant impact on the activity of enzymes such as urease, catalase, fluorescein diacetate hydrolase (FDAse), and phenol oxidase. Changes in these enzymes can affect soil fertility [113,114]. Moreover, Huang et al. [112] observed that, after 15 days, MPs (2000 particles/kg soil) considerably boosted the urease and catalase activities in soil, but invertase activities showed no detectable change.

Agroecosystems may be significantly endangered by MPs due to their greater effects on the dynamics of soil carbon and nitrogen cycling, according to a study by Meng et al. [115]. The addition of MPs increased enzymatic activity; activated pools of organic C, N, and P; and was favorable for the accumulation of dissolved organic C, N, and P [116]. Additionally, according to Liu et al. [111], it is presumable that adding MPs to soil greatly increases its levels of nitrogen. Additionally, Feng et al. [117] showed that MPs reduced the amount of N and P readily available in the soil but changed the bioavailability of Pb and Zn, as well as the activity of soil enzymes, differently. MPs can influence the carbon cycle in a variety of ways; for instance, they can be carbon themselves and impact plant growth and soil microbial activity, or through litter breakdown [118]. Soil bulk density is a key parameter for the extrapolation of soil carbon storage, which is greatly influenced by MPs, and the presence of MPs may result in the underestimation of soil carbon storage [119]. Furthermore, MP-C may be misidentified as a large anthropogenic component of the soil organic carbon pool since MPs contain high carbon polymers (for example, polystyrene or polyethylene are almost 90% carbon) [120]. The addition of MPs to the soil according to Feng et al. [117] changed the soil's pH and dissolved organic carbon levels in a variety of ways. Additionally, different studies [111,121–123] showed that the accumulation of high-molecular-weight humic materials resulting from MPs play a role in enhancing soil quality, as humic materials can enhance soil stability, water-holding capacity, and nutrient availability, among other things. In contrast, Yang et al. [124] discovered that, under low MP dosage, as opposed to high MP dosage, the breakdown of soil organic matter (SOM) was significantly higher. In addition, the influence of MPs on the mineralization of exogenous C substrate was minimal (glucose or straw).

Accumulated plastic particles in plants may have a blockage effect on cell connections or cell wall pores, affecting the plant's ability to absorb and transport nutrients [125]. Bosker et al. [126] showed that the germination rate of seeds exposed to 4800 nm MPs decreased from 78% in the control to 17% in the maximum exposure. According to Qi et al. [127], wheat biomass was reduced after receiving a 1% weight-by-weight treatment with several polymers. In contrast, Lozano et al. [100] found that all MP morphologies (fibers, films, foams, and pieces) increased shoot and root masses. Moreover, Machado et al. [70] found that some MPs enhanced the root biomass and form of spring onion roots (*Alium fistulosum*). In another study, Sourkova et al. [78] found that an increase in NP concentrations (25%, 50%, and 75%) due to tire abrasion had a significant impact on phytotoxicity, with higher-NP substrates inhibiting the root growth of *Lepidium sativum* L. ($IR_{LS}$) by 86%, 80% and 62%, respectively, and *Sinapis alba* L. ($IR_{SIA}$) by 88%, 84% and 70%, respectively, compared with the control samples. Pradel et al. [68] suggested that the main cause of MP effects on roots could be changes in soil microbial activity and pore connectivity, which affect water evaporation.

Generally, MPs can alter a variety of soil properties and impose certain selection pressures on soil microorganisms, which alter the community's structure and diversity and have an evolutionary impact [120]. Studies have revealed a direct relationship between soil properties, such as nutrient level and microbial activity [128–130]. Changes to the soil's physical environment, especially soil aggregation, which has been found to incorporate linear microfibers, are anticipated to have a different impact on microbial evolution than soil that is not microfiber-structured [109]. Lehmann et al. [131] also showed that when soil bacteria were alive compared with when they were sterile, aggregates generated from nothing were significantly greater; however, this advantageous effect was lost when soil microbes were exposed to microfibers. Rillig and Lehmann [118] predicted that the relative distribution of aerobic and anaerobic bacteria would change due to changes in

soil porosity and moisture caused by MPs. Veresoglou et al. [132] also discussed the potential loss incurred in the microenvironment and the extinction of native microorganisms as a result of MP-induced changes to pore spaces. Additionally, in another study by Veresoglou et al. [120], it was found that the addition of MPs significantly impaired the ability of the microbial community to organize. Fei et al. [133] also found that the relative abundance of Burkholderiaceae families significantly increased following MP addition ($p < 0.05$), indicating that the presence of bacteria is associated with MP-catalyzed nitrogen fixation. Additionally, there was a significant decrease ($p < 0.05$) in Sphingomonadaceae and Xanthobacteraceae with the addition of 5% polyvinyl chloride (PVC) and 1% PE MPs, respectively, indicating that MPs may impede the biodegradation of bioformers in the soil. McGonigle et al. [134] found that MPs changed soil fungi at different levels.

Recent investigations have discovered that certain soil micro- and macroorganisms can feed MPs [135,136] and that MPs can be ingested by animals [137], which can have negative health effects. Several academic studies have demonstrated that soil organisms, such as earthworms, absorb and excrete NP, along with other natural soil particles [101]. Ingesting MPs from microorganisms may cause false saturation, which slows the consumption of carbon biomass and eventually results in energy depletion, stunted growth, and even mortality [138,139] (Figure 7).

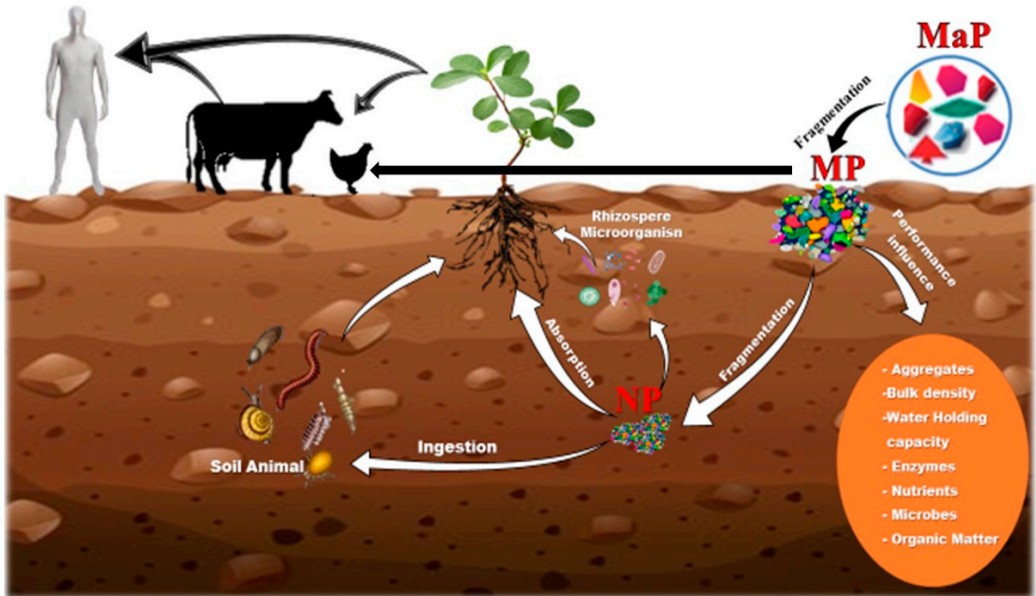

**Figure 7.** Possible pathways of MP and NP uptake in soil, plant, animal, or human.

### 3.3. Impact of MP and NP on Human Health

MPs are among the most harmful environmental pollutants because they endanger both human health and the health of land and marine ecosystems. Depending on their size and type, NPs can enter the seed, root, culm, leaf, and fruit plant cells [140]. In addition, like nutrients, NP is also absorbed in plant roots and is subsequently transported to the tissues through endocytosis, accumulated in plant tissues, and will ultimately be consumed by consumers [141,142]. The most serious issue is that humans and other species may absorb plastic in the form of NP via food, according to a summary by Chen and Wang [143]. Every day, NP particles enter the human body through water, air, and foods like fruits and vegetables [144–146] (Figure 7). MPs may cause particle and chemical toxicity when inhaled, as well as various respiratory symptoms. Moreover, reports on occupational illnesses in industry workers also point to the harmful effects of MPs [139]. Velzeboer et al. [147] claimed that NPs have a large specific surface area and higher sorption affinities for hazardous compounds, which may ultimately lead to enhanced toxicity.

Ibrahim et al. [148] provided evidence that MPs are able to travel into the human colon. NPs found in the placentas of pregnant women have also been reported by Ragusa et al. [149] and Braun et al. [150]. Ballesteros et al. [151] exposed whole-blood samples from different donors to different concentrations of MPs–NPs to assess genotoxic and immunomodulatory effects in human white blood cells. This led to a significant increase in DNA damage in monocytes and polymorphonuclear neutrophil immune cells (PMNs) as well as changes in the whole-blood secretome. The study by Leslie et al. [152] confirmed the presence of MPs–NPs in human blood. Bartucci et al. [153] studied the uptake of MPs–NPs in liver, lung, and kidney slices. Until now, researchers have not sufficiently investigated whether and via which pathways MPs, but especially NPs, enter plants, soils, and particularly animal and human bodies or cells, and what effects they have on human health [135,153]. But all of these findings raise serious concerns.

## 4. Conventional Disposal Processes of Plastic Mulch

Plastic films are frequently burned or dumped in landfills due to the significant expenses associated with the collection, disposal, and recycling of films. According to OECD [59], in 2019, 49% of plastic waste was globally disposed of in landfills [92]. Following such procedures, hazardous chemicals are released into the air and soil [122,154,155]. Due to this fact, there is a significant amount of plastic garbage that is still present in the environment, which causes pollution and endangers the natural system [92]. Each European nation has a different set of regulations on landfilling. Less than 10% of plastic trash from agricultural and non-agricultural sources is disposed of in landfills in central Europe and Scandinavia (with the exception of Finland). However, estimates state that, in Spain and the bulk of eastern and southern European countries, more than 50% of their plastic waste is from trash [156]. The industrial incineration of plastics is a practical alternative, notably for energy production, as landfill space is becoming smaller, and concerns about the disposal of tainted plastic grow [52]. Through the incineration of plastic films, there is an ability to recover energy. However, there are obstacles due to plastic soiling, because most power plants and incinerators are not able to incinerate plastic that has been covered in dirt and debris [66,157]. Increasing environmental concerns about the disposal of plastic mulch through incineration and dumping in landfills or via export in foreign countries have already led to restrictions in some areas. Consequently, the agricultural industry needs a method for the disposal of plastic waste that is environmentally friendly [122,154]. Without the effective recovery and recycling of mulch films, the residual plastic film poses contamination risks to agricultural systems and natural environments [95].

## 5. Recycling of Plastic Mulch

Due to the environmental risks associated with the accumulation of plastic films in the environment, especially in agricultural systems, it should be collected and disposed of through recycling companies. However, the options for recycling used PE mulch after its use are limited [158,159]. Questions have started to arise over common agricultural practices such as the disposal of plastic water pipes, greenhouse covers, and mulch [31,99,160]. Only 9% of worldwide plastic waste was recycled in 2019, and 19% of it was burned to make electricity [92]. OECD [59] expects global plastic production to continue to increase, reaching a total of 25,000 million metric tons by 2050. At the same time, the quantities of plastics that are discarded, incinerated, and recycled will increase. First, from the year 2050, the amount of discarded waste will decrease (Figure 8). According to a 2016 World Economic Forum study, just 14% of packaging garbage is recycled, and more than 80% of it is thrown away or dumped as litter, 40% of which is landfilled, 14% is burned, and 32% is released into the environment [161]. Mulch films account for only 12% of the agroplastic mass on the EU market [162]. According to Plastics Europe [38], close to 29.5 million tons (34,6%) of post-consumer plastic garbage were gathered and transported to recycling facilities both inside and outside of Europe (Figure 9).

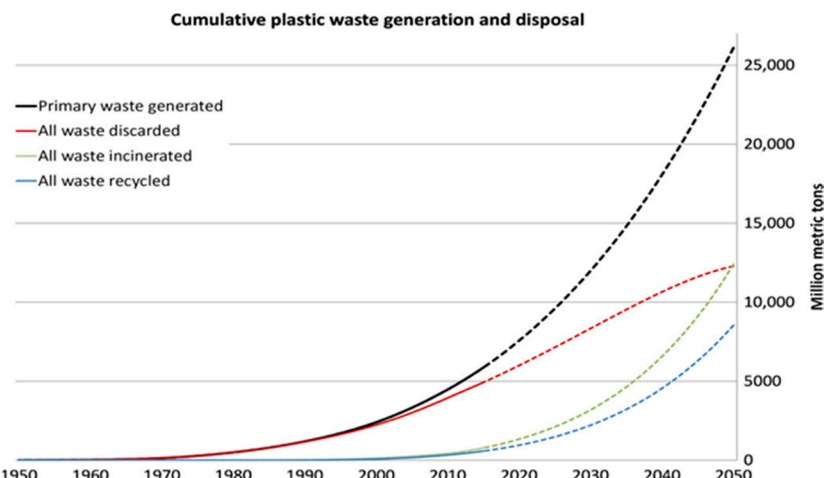

**Figure 8.** Cumulative worldwide plastic waste generation and disposal (in million metric tons). Solid lines show historical data from 1950 to 2015; dashed lines show projections of historical trends to 2050 [92].

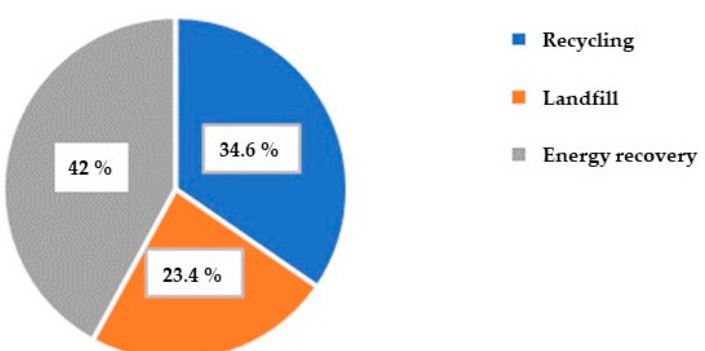

**Figure 9.** Disposal of plastic waste in EU in 2020 [38].

As the problem of plastic waste constitutes one of the most important environmental risks around the world, finding solutions and implementing projects for recycling this waste is imperative. Therefore, many projects and programs for recycling this waste have been established in many sectors. In the United States of America, Florida Agricultural Plastic Recyclers (FLAG) and Cornell University's Recycling Agricultural Plastics Program (RAPP) have developed methods for effectively collecting and recycling used agricultural plastic [63]. The outcome has been the collection of more than 1.2 million pounds of unwanted agricultural plastic that would have otherwise been abandoned on fields, burned in open fires, or buried in landfills over the two years that RAPP has had plastic compaction equipment funded by a prior contract with the department. Through cooperative agreements, Soil and Water Conservation Districts (SWCD) and the government jointly owned and managed these balers. Some of the gathered plastics have been turned into plastic oil, drainage tile, and sidewalk pavers [163]. These 1.2 million pounds of plastics, according to RAPP staff estimates, account for around 4% of all the agricultural plastic used in New York State during this time [163]. In 2021, Irish farmers recycled farm plastics at a record-breaking rate of 90%, recycling enough plastics to make 18 million silage bales. Over 200 bring centers hosted by the Irish Farm Films Producers Group [164] at locations such as markets, coops, and agri-merchant premises collected the vast majority of the plastic waste during the summer. This helped Ireland's efforts to promote a circular economy because more than a third of the materials collected were sent to Irish recyclers to be turned into several new products.

However, the contamination of plastics is one of the main problems in recycling [165,166], as many films are highly contaminated with soil, organic materials, or water, as well as

agrochemicals. For example, white asparagus ridges are covered with black plastic films during harvest to simplify the harvest process. These films have pockets on both sides, which are filled with sand. After about 8 years, the films must be replaced and disposed of. They can weigh 90% less when the sand is removed. In a study by Geyer and Salama [167], a process is described for cleaning the side pockets of asparagus films without producing MPs in the shaken-off sand. By cleaning the film before the recycling process, large amounts of dirt, sand, and water remain on the farm and can be spread on the field again, transport costs are significantly reduced due to less volume and mass, and the recycling process is simplified and becomes less expensive. Most recycling companies will only accept these films if the sand-filled pockets are emptied first.

The recovery of used agricultural films has decreased in part as a result of China's 2013 "Green Fence" initiative, which set rigorous criteria for the quality of recyclables deemed acceptable, according to a research study by Moore Recycling Associates Inc. Plastic film will not be accepted for recycling if there is contamination above 5% of the mass [168].

The German organization ERDE, which received funding from the film industry, worked to collect and recycle a total of 26,910 tons of agricultural films in 2021. These efforts took place at 543 fixed collection stations and in 1936 mobile collections [169]. In 2022, ERDE collected and recycled 68.7% of the silage and stretch films sold in Germany, which amounts to around 34,000 tons annually, in addition to adding mulch film in the national collection scheme [170].

Figure 10 illustrates different residue materials on used mulch films of different horticultural crops in France between 2016 and 2019 [171]. For example, on average, mulch films removed from an asparagus field contained 3% organic matter (as a percentage of the total weight of used mulch film), 10% water, 15% loam, and 50% sand. To remove and clean mulch films, an innovative technology was developed in the French project "Recyclage Agriculture et Films Usages" (RAFU-II). A machine picks up the mulch film; removes dirt and plant debris by shaking, brushing, and blowing; and then winds it up. Depending on the vegetable crop, up to 69% of the residue materials could be removed using this technique. This can also significantly reduce transport costs to the recycling company due to less volume and mass of the mulch film. By reducing the amount of dirt from 70% to 30%, the transport costs to the recycling company for 1000 tons of used mulch film could be reduced from EUR 143,000 to EUR 55,000. The costs for collection and recycling must be borne by recycling companies [171]. To cover all the financial requirements, the ecocontribution of a new film of EUR400/t to EUR500/t would have to be taken into consideration, increasing the cost of the new product by the same amount [171].

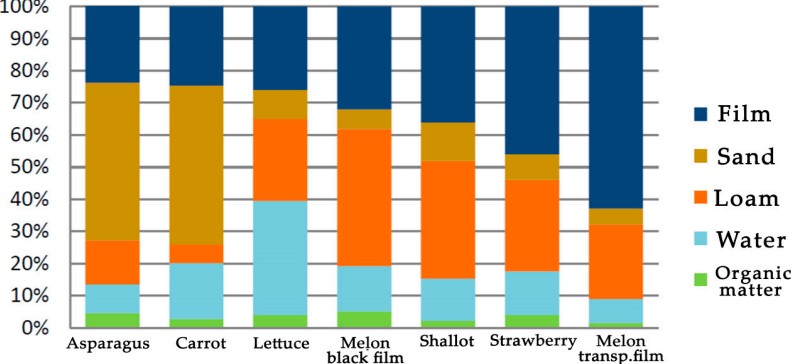

**Figure 10.** The average proportion of different residue materials in used mulch films of different horticultural crops in France between 2016 and 2019 [171].

## 6. Alternatives to Plastic Mulch

The long-term development of substitutes for PE mulch is needed to reduce the risk of environmental pollution from plastic residues and their harmful effects on the environment [47,172].

Biodegradable films were produced and brought to market because it is difficult to recycle traditional PE mulch films after use, especially the extremely thin films used in Asia. These biodegradable plastic mulches behave similarly to PE mulches but are much more environmentally friendly and do not need to be removed and disposed of at the end of cultivation [173]. However, many strawberry growers in different regions of the USA are very reluctant to use such films [174]. A polymer's biodegradability refers to its capacity to break down into simpler compounds, including water, carbon dioxide, methane, and basic elements, in the soil [175,176]. The following criteria must be met by mulch films: (1) the biodegradation of at least 90% after 24 months and (2) the lack of heavy metals and the absence of ecotoxicological effects. Although studies on the toxic effect of the decomposition of biodegradable mulch are not sufficient, some views still warn against its widespread dispersion without confirming the safety of the decomposition process [177]. However, the USDA National Organic Program (NOP) has adopted the same specifications for mulch films permitted for organic crop production [178,179]. But at this time, no biodegradable mulch is allowed for use on organic farms. None of the commercially available biodegradable mulches have been proven to meet the requirements of the NOP standards [179]. However, consecutive applications of soil-biodegradable mulch did not have a negative impact on soil quality compared with applications of PE mulch, according to the findings of a 4-year study headed by Sintim et al. [180]. According to estimates from 2022, the market for soil-biodegradable mulches was worth USD 62.7 million. By 2026, however, that market is expected to have grown to USD 83.6 million [181]. After being used, biodegradable films, which are often made of bio-based materials, can be crushed and buried in the soil with plant detritus so that bacteria, fungi, and algae [182,183] can degrade them [184]. These environmentally safe polymers degrade after many months, and the environment's moisture content, oxygen availability, temperature, kind biomass, humic matter, and the number of microorganisms and enzymes (bacteria, fungus), as well as the amount of salt present, all affect the amount and rate of biodegradation [185,186]. According to a qualitative study by Goldberger et al. [187] involving crop growers, agricultural extension agents, agricultural input suppliers, mulch manufacturers, and other stakeholders, the main adoption barriers were a lack of sufficient knowledge, high costs, and unpredictability in the breakdown. Kasirajan and Ngouajio [188] also bolster these findings. Temperature, humidity, fertilizer, and the substance used to make biodegradable mulch are all related to how quickly things decompose [189,190]. The soil microbial biomass and glucosidase activity responded favorably to the burial of biodegradable mulches for 18 months [24]. It is not necessary to remove the biodegradable plastic mulch at the end of the growth season. This lowers the cost of transporting, recycling, and salvaging films. The potential labor savings from its use are crucial in farmers' assessments of the economic sustainability of switching from PE mulch to biodegradable plastic mulch because biodegradable plastic mulch is more expensive than PE mulch [64]. Biodegradable polymers are not commonly used because of their poor performance and increased costs relative to traditional plastic mulch [191].

Polylactic acid (PLA) is now the leading example of a synthetic biopolymer that is biodegradable, renewable, and recyclable. Chemical synthesis or enzymatic polymerization can be used to produce PLA [191]. The benefits of PLA polymers also include their ability to be altered to achieve desired mechanical and physical properties that are comparable to those of synthetic polymers, in addition to the significant energy savings and carbon dioxide gas exhaustion achieved during their production, as compared to that of synthetic fibers [192,193]. An effective substitute for polypropylene (PP), PE, and polyethylene terephthalate (PET) fiber is PLA fiber [192].

Another choice as an alternative to plastic mulch is biodegradable paper mulch film. It is environmentally friendly and provides a new direction for developing degradable plastic film, which is widely used in Japan [194]. Güzel [195] found that, in terms of preventing weed growth and managing topsoil temperature and moisture, paper mulches are comparable to traditional opaque PE plastic mulch films and biodegradable plastic mulch films.

The main substances in paper mulch are cellulose, starch, and other renewable materials, which enable paper mulch to achieve pollution-free soil self-degradation, improve soil temperature and water content, as well as encourage the growth of soil microorganisms, reduce the impact of weeds, diseases, and insect pests on crops, and ultimately increase crop yield [14,196]. Biodegradable paper mulch films, which are entirely biodegradable and may be used as a better green mulching alternative in agricultural mulching, have been shown in realistic field studies to increase soil qualities and crop productivity [197,198]. Li et al. [199,200] used ZnO and $SiO_{220}$ as basic materials to create a biodegradable super-hydrophobic paper mulch layer utilizing a straightforward coating technique. In order to develop a fresh approach to enhancing the water resistance and weathering resistance of this biodegradable paper mulch film, they examined the aging performance, frictional wear study, and low-temperature environment. Different climatic parameters such as rainfall and other external factors also affect the impact of degradable paper film on soil temperature, soil water-holding capacity, the control of weed diseases and insect pests, and self-degradation [201]. Although biodegradable paper mulch film's mechanical properties, moisture retention performance, and warming performance were inferior to those of conventional plastic film, it outperformed the latter in terms of antiaging performance, soil oxygen content, heat preservation, and water storage capacity in the middle and late stages of crop development [194]. Similar results were found by Güzel [195], who found that paper mulches produced from mechanical or chemical pulps were not always strong enough to withstand the strain caused by laying. The strength of the paper could be increased through specific technical processes in the production such as Clupak and creping. Plant fibers are essentially divided into five categories based on their botanical origin: bast (kenaf, hemp, ramie, flax, jute, banana, etc.); leaf (sisal, agave, abaca, pineapple leaf (PALF), etc.); seed (cotton, kapok, soya, rice hulls, etc.); fruit (coir, oil palm, etc.); stalk; other origins (rice, wheat, maize, rye, oats, etc.) [202–204]. The kind and part of the plant fibers, cellulose, hemicelluloses, lignin, pectin, and wax constitute the majority of plant fibers [205]. The quantities of these components vary. The top priority of the Food and Agriculture Organization (FAO) is to utilize natural fibers in the development of new materials since doing so will improve global agriculture sectors' productivity and sustainability. Natural fibers are readily available, renewable, and biodegradable, and carbon dioxide remains in the cycle. They have high modulus, low density, high elongation, and high elasticity, as well as excellent mechanical strength and moisture absorption. Because of the relatively simple manufacture and low processing costs, they are a significant economic priority for developing countries [193,206–208]. Due to their low resistance to microorganisms and pests, high moisture sensitivity, and moderate heat stability, natural fibers have a shorter lifespan than synthetic materials [175,209].

Due to long-term environmental concerns regarding the persistence of PE mulch films in the environment and the use of heavy metals (e.g., Co) in oxo-degradable films, biodegradable, bio-based, and compostable polymers are increasingly sought as replacements. Oxo-degradable plastics are plastics that have the desired property of rapid fragmentation after use. However, oxo-degradation is not to be confused with biodegradability. Fragmentable plastic is a type of plastic that is degradable but cannot be fully decomposed by microorganisms or compostable or biodegradable according to the applicable standards for organic recycling or biodegradability of plastics and packaging [210].

A solution has been proposed to the issue of the high application cost of biodegradable mulch: the development of a sprayable biodegradable mulch. Several studies [211–214] and the EU Project BIO.CO.AGRI [215] revealed several sprayable biodegradable soil mulch materials using natural polymers such as starch, cellulose, chitosan, alginate, and glucomannan. Using natural polymers from seaweed and crab shells, researchers from the Institute of Chemistry and Technology of Polymers (ICTP) group have also developed a sprayable biodegradable composition [216]. Research has also been conducted on sprayable protein-based hydrolyzed films [217,218]. Adhikari et al. [219] indicated that spraying solutions are a typical agricultural practice, so the use of sprayable mulch should be straightfor-

ward. Employing a spray cannon to apply mulch in a greenhouse environment requires far less labor than using premade film, which must be measured, cut, and placed [220]. A sprayable biodegradable mulch differs from a pre-formed biodegradable plastic mulch in its interaction with the soil. A sprayable biodegradable mulch will form strong physical, and potentially chemical, interactions with the soil, and will draw its strength from its interaction with the soil surface, as opposed to only from interaction with itself. This is an important difference as it could cause differences in degradation and efficacy [221]. According to a study by Niemann et al. [222] on iceberg lettuce plants in Australia, fewer aphids occurred at the first time of infestation due to the black-gray spray coating. The study's objective was to reduce the early aphid invasion (Macrosiphum euphorbiae, Nasonovia ribisnigri). Furthermore, the spray treatment had no detrimental effects on the quality of the lettuce heads. Giaccone et al. [223] showed that even in situations of extreme weed infestation, spray mulch efficiently suppressed weeds for more than two months following application. Immirzia et al. [220] developed and tested a biodegradable sodium alginate water-based coating that may be sprayed. The coating maintained its ability to suppress weeds, and biodegradation studies revealed that the spray coating samples biodegraded by 65% after 6 months into the soil. They also discovered that the influence on the soil's heat balance fluctuated without predictable trends. In preliminary investigations, it was found that spray-on-mulch (SOM), which uses a slurry mix of recycled newspaper waste and chopped cereal straw at a rate of 3 kg/m$^2$ (dry weight (DW)), was effective in suppressing the establishment of most annual weeds but not all of them. SOM significantly reduced summer and wintertime temperatures, conserved moisture in the summer, and had no effects on plant nutrient status [224]. The results of Braunack et al. [225] demonstrated that crop emergence reduced with the increase in the application rate (i.e., control, 0.25 kg/m$^2$, 0.5 kg/m$^2$, 1 kg/m$^2$) and was proportional to the application rate. Additionally, the sprayable biodegradable polymer membrane (SBPM) was most effective at 0.5 kg/m$^2$ and 150 mm width and inhibited weed growth similarly to standard mulch film [225]. Remmele [226] investigated sprayable mulch materials with biogenic origins that decompose once vegetable life is over in field cucumbers, zucchini, and carrots in ridge culture, in addition to lettuce and kohlrabi row crops. However, like organic mulch, the sprayable biodegradable mulch has the drawback of requiring a disproportionately large amount of material.

Organic mulches are made from plant and animal products such as straw, hay, peanut hulls, leaf mold, compost, sawdust, wood chips, shavings, and animal manures [227]. Employing organic mulches will allow for weed suppression at acceptable levels [2,228–230]. It has been demonstrated that organic mulches can have a variety of effects on weeds. To interfere with the circumstances necessary for weed growth, they can, for instance, block visible light and lower soil temperature, and thus generate a particular microclimate [231–233]. Additionally, organic mulches can affect weed germination and emergence by releasing allelopathic substances or by altering the chemical properties of the soil, such as the soil's C:N ratio [234–236]. The efficiency of organic mulches in minimizing soil and water losses under various climatic situations around the world has been demonstrated in relevant papers from Asia [237], Europe [238,239], Africa [240], and America [241]. By lowering evaporation, organic mulches can enhance soil water content and keep soil temperatures more stable [242,243]. As they are added to the soil toward the conclusion of the growing season, organic mulches might increase the amount of organic matter in the soil [224,244,245]. Additionally, organic mulches are effective at reducing nitrate leaching, enhancing the physical characteristics of the soil, preventing erosion, and boosting soil biological activity [20,227]. The main drawbacks of organic coverings are their high cost, huge quantity requirements, lack of premature effect, and lack of flexibility compared with plastic films. Spreading and moving organic mulch during crop cultivation is difficult and requires a considerable amount of human labor [27]. This increases costs, complicates logistics, and limits the use of organic mulch for growing horticultural crops.

## 7. Conclusions

Plastic film mulching is an important procedure for improving crop growth and yields in modern horticulture and agriculture by conserving soil water, regulating soil temperature, and suppressing weeds to avoid pesticide contamination. One of the most important benefits of PE plastic mulch is its availability, cheapness, and flexibility compared with other materials. Regardless of the benefits, the fact that PE mulch is produced from non-renewable petroleum-based polymers, and the need for its disposal after use, makes the use of this material a concern. Until now, PE mulch is frequently disposed of in landfills or through field incineration, which leads to several serious environmental issues.

In addition, PE mulch is an important source of MP soil contamination. The degradation and fragmentation of PE mulch due to climatic factors or mechanical processes on mulch film in the field release MP particles into the soil, which negatively affects soil structure, soil density, porosity, evapotranspiration, and the water-holding capacity of the soil. It also affects the enzymatic and microbial activity in the soil. In addition, MPs affect plant growth and productivity. Further fragmentation of MPs results in NP formation. Especially dramatic is its accumulation in plant tissues and animal and human bodies with the result of possibly negatively affecting human health.

The amounts of MP remaining in the soil vary depending on the thickness of the PE mulch—films that are too thin tear and the remains stay in the soil—and the intensity of use, as the amounts of MPs accumulate over the years, causing contamination. It is therefore necessary to collect PE mulch as completely as possible from the field at the end of the season. However, plastic mulch dirty with soil and plant residues hinders the recycling process and also increases the costs of transportation to recycling companies, as well as recycling costs. Various initiatives have been implemented and technical processes have been developed to facilitate the collection, cleaning, and subsequent recycling of these films. However, these initiatives need to be significantly intensified to recycle even more plastic. Overall, agricultural advancements for reducing plastic waste in developed countries are on a good path. In contrast, major efforts are still needed in developing countries.

Biodegradable plastic mulches and paper mulches are some of the alternatives and have similar benefits to PE plastic mulch, although they are more environmentally friendly. However, due to limited durability and higher material costs, biodegradable polymers are not widely used. The advantage is that biodegradable plastic and paper mulch do not need to be removed at the end of the season. Another option is the use of organic mulches like fibers. They are readily available, renewable, and biodegradable, and emit no carbon dioxide. They have high modulus, low density, high elongation, and high elasticity, as well as excellent mechanical strength and moisture absorption. The disadvantages of organic mulches compared with plastic mulch are the relatively large amounts of material required and the high costs for material, labor, and transport; thus, farmers do not widely adopt them.

All in all, it is difficult to compare the different mulching methods. They differ in the labor required for application; their cost; the quantities that need to be applied; and their effect on weeds, temperature, nitrogen fixation, and plant growth, but also their $CO_2$ emissions. From a cost perspective, PE mulch is the least expensive. However, in terms of disposal, biodegradable films and paper mulches that are left on the field and incorporated may have advantages. In terms of $CO_2$ emissions, however, all biodegradable mulch materials have an advantage. They are produced from renewable resources and the carbon thus remains in the cycle.

**Author Contributions:** Conceptualization, M.G.; methodology, M.G. and K.S.; resources, M.G.; writing—original draft preparation, K.S. and M.G.; writing—review and editing, M.G. and K.S.; supervision, M.G.; project administration, M.G. All authors have read and agreed to the published version of the manuscript.

**Funding:** The project is supported by funds from the German Government's Special Purpose Fund held at Landwirtschaftliche Rentenbank. The publication of this article is funded by the Open Access Fund of the Leibniz Association.

**Data Availability Statement:** All references used are publicly available.

**Conflicts of Interest:** The authors declare no conflict of interest.

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
