# Peer review of "Plastic Mulch Films in Agriculture: Their Use, Environmental Problems, Recycling and Alternatives"

_environments, doi:10.3390/environments10100179_

Round 1

Reviewer 1 Report

General comment

In their paper “Horticultural Plastic Mulch -The Impacts and the Environmental disposal” the authors aim “to highlight the most of plastic mulch utilizes and its damages to the environment in general and the soil in particular. In addition to the conventional methods of disposal, methods of collecting and preparing the mulch for recycling to prevent environmental pollution as well as plastic mulch alternatives are discussed.” (lines 17-21). The topic is in the scope of the journal. The idea is good and such a review would bring value added to the literature, mainly, if it reviews information on the conventional methods of disposal of plastic mulch film, of collecting plastic mulch film on preparing plastic mulch film for recycling. Given the raising number of studies on plastic (MaP, MP, NP) including review studies, it is a big challenge provide an exhaustive and updated literature review. However, I have some comments, which I hope can help the authors, to improve the literature review.

Major comments

Major comment 1

In many places the text does not inform about the reference time or region. It is not clear if the statement refer to “world-wide”, or specific countries or regions. This information is of high interest and relevance. It could be also of interest, what is the current legal status of plastic waste disposal in the countries. In some European countries, the disposal of plastic waste by landfill, might not be an option anymore.

Major comment 2

The article does not fully provide, what the abstract announces. The review on impacts, and alternative mulch options are elaborated but on impacts and alternative mulch options, already review article exists focussing only on these topics. The information on disposal seem to refer to plastic waste in general and not to plastic mulch film in particular. The information on collecting and recycling is better elaborated and might be the original contribution of the paper.

Major comment 3

The text seems not be very carefully elaborated. Most of the figure seem to be directly copied from the original sources and not formatted in an article specific layout. Such an original presentation, can be OK in review articles, but leaves an impression on the originality of the work. For at least one reference the presented information might not be the best one (see minor comments).

Major comment 4

The manuscript would benefit from a section presenting existing reviews on plastic mulch film and demonstrating which gap the current review is filling.

Major comment 6

Overview tables could proved an easier overview than only presentation in text, e.g., country wise presentation of recycling programs in comparison, with indicating the information of interest (starting year, collection rate, plastic types collected, ….).

Major comment 7

The title “Horticultural Plastic Mulch -The Impacts and the Environmental disposal” Is in my opinion not to be very clear. Does “The impacts” refer to the positive agricultural impacts and to the negative impacts from plastic emissions? Does “environmental disposal” mean littering or landfilling, does it include also plastic burning and recycling?

Minor Comments

Line 27-36:

The authors start the introduction, with “Mulching is a crucial horticultural technique to increase crop productivity in areas with minimal water input”, which could create the impression that mulching is primarily applied to safe water. This might apply in water scarce regions more, but on a broader sense, in regions (and times) without water scarcity, mulching is applied also to: prevent weet growth, to maintain soil structure, etc. The authors could maybe compare with other reviews on mulch film. The purpose of mulch film application is in most of the reviews described like the authors do in the abstract: as multi functional use.

Line 35

Please, check the journals’ formatting of citations. The narrative quotation “[14,15] observed that ….” Reads somehow bizar. The authors might want to check if not the following format would be in line with the journal and easier to read: Acharya et al. [14] Muhammad et al. [15] observed that ….”

Also check the bibliographic information for [15].

Line 57-58: “In Europe, mulch film has a thickness usually between 15 and 80 µm covering annually or perennially the soil in crop production tems [24].” Source [24] might not be the best suitable to reference the mulch film thickness in Europe, since the study is on Germany and focussing on estimating film quantities, emissions and abatement costs. The updated version of the preprint also contains some references in the Discussion section (hal-03779834 , version 2 (18-08-2023) ).

Line 62: Check wording: In terms of Europe,” versus “In Europe, “

Line 62-63 (an other ocasions)

Check the usage of the decimal sign “,” versus “.”.

Line 107-108:

“Due to worries about the disposal of contaminated plastic and the decreasing amount of land accessible for landfilling, industrial burning of plastics is a viable alternative.” Thus, it could be indicated, where, mulch film is disposed as landfill. Also the quoted source [50] is from 1993, thus, maybe could be questioned as old, thought the content might not have changed.

Line 202 (and other  occasions)

Check the number of decimals for reporting, if reporting 2 decimals are required useful: “decreased from 58.03% in the treatment without residual film to 57.36% in biodegradable”

Line 310

The authors seem to present “Figure 6. Impact of MPs on soil, plant and human.” As the only graphical element in the article. At least for the other figures the sources are indicated and for some (e.g., Fig 3) the graphs seemed to be copied directly. The direct copying might explain the different quality of the graphs, which could create the impression, that the authors copied graphs from different sources together. The problem with the seem-to-be original figure 6 is, that, there are flows missing from MP to organism. According to Figure 6, the impact chain is: microplastic à microplastics à nanoplastics à organism. According to other authors (e.g.; Ng et al 2018) also microplastics can be ingested by animals and create damages. The Fig 6 suggests, that the impacts occur only with the breakdown to nano plastics. Note, that MP are suspected to adsorb and transport active substances (e.g., agrochemicals), which can be ingested by animals.

Literature

Ng EL, Huerta Lwanga E, Eldridge SM, Johnston P, Hu HW, Geissen V, Chen D. An overview of microplastic and nanoplastic pollution in agroecosystems. Sci Total Environ. 2018 Jun 15;627:1377-1388. doi: 10.1016/j.scitotenv.2018.01.341. Epub 2018 Feb 20. PMID: 30857101.

Line 311

The section header is titled as “3.3 Imapct on Humand Health”, however, the section does not describe the impacts. It describes the suggestion by different studies, that big quantities of NP are ingested by humans. However, the impact of nano plastic is not described, and according to my knowledge it has not been scientifically proven.

Line 550-551

Check spelling “Asia [210], Europe [211,212], Afric [213], and America [214].”

Line 413

Consider also quoting: https://www.sciencedirect.com/science/article/pii/S2352186422001110

Lines 753-574

The authors conclude that “In addition, polyethylene mulch is one of the most important sources of MP soil contamination.” This conclusion seems to be general, but might not apply to all regions. For example, Brandes et al (2020) identify sewage sludge and compost as significantly higher emittant for MP into soils at least for Germany. Brandes et al (2021) (https://iopscience.iop.org/article/10.1088/1748-9326/ac21e6/meta )

Lines 323-343

The header “4. Plastic mulch disposal processes” indicates that the reader is informed about the different ways of how plastic mulch is disposed. The text however, refers in general to the disposal of plastic waste in a general manner. And also the key reference by Geyer et al (2015) on the quantity of plastic ever made, rather informs on plastic waste and recycling rates for plastic in general, than for plastic mulch films

Lines 324-326

Plastic films are frequently burned or dumped in landfills due to the significant expenses associated with collecting, disposing, and recycling films. 79% of plastic waste in 2015 was either disposed of in a landfill [78].” To which region do these figures refer? And do the 79% of plastic waste refer to the disposal of mulch film?

Lines 417-419

“Biodegradable films have been produced and put on the market due to the difficulty in recovering traditional polyethylene mulching film after usage, especially the incredibly thin films used in Asia.”

Check expression “put on the market”

The text might benefit from an English editing service.

Author Response

Dear reviewer 1

I want to send you  3 files:

The manuscript with and without comments and the reviewer comments

Reviewer 2 Report

Please see attached document (PDF).

Overall, I can understand the paper. A moderate editing is required to improve the clarity of some sentences. 

Author Response

Dear reviewer 2

I wanted to send you the manuscript with and without comments and the answered comments of reviewer 2.

But Iam not sure if all files will be send 

Round 2

Reviewer 1 Report

General comment

The authors have addressed most of my comment. However, for some part, the text could benefit from some minor revisions. Particularly the quatations could be checked carefully through the whole text. I indicated some issues, but there might be more within the text.

Minor comments

The authors might want to check some issues in the text, particularly with respect to the citation.

Line 40

„Acharya eat al [20], Muhammad et al [21] observed“ à Acharya et al [20] and Muhammad et al [21] observed

Line 47 52

Gao, Yan et al [24] and Liu, Bu et al [26] à Liu  et al [26] ###no need to quote more than the lead author, since the number identifies the reference###

Line 137

First should it be: [45] [68] à [45, 68]  the reference [68] appears in the text before reference [60] (in

Line 364 (Figure 6) and athors answer: “We drew an arrow from MP to plants and animals/humans but we are not sure because cell membranes can be penetrated only by NP.”

Comment: the figure is titled: “Figure 6. Impact of MPs on soil, plant and human.” Thus, according to the title the main focus is on the impact of MP. MP might not pass the plants membrane, but can be absorbed by animals. The impact for health is not clarified yet. If the authors have doubts it would be good to compare with other studies. There is a big number of MP impacts studies and reviews available in the literature.

Line 549 “Güzel, E [196]“ à „Güzel [196]“

Line 564 comma at the end of sentence

Line 596-600

The long-list-quoting could be avoided by changing:

“In addition, accord- ing to Avella et al. [213], Mormile et al. [214], Malinconico et al. [215], Schettini et al. [216] and the EU Project BIO.CO.AGRI [217] ..”

TO

“In addition, according to different studies [e.g., 213-216] and the EU Project BIO.CO.AGRI [217] …”

Line 694-699

“Other options are the use of organic mulches like fibers. They are readily …

material, labor and transport and are thus little accepted by the farms.”

Comment: The authors might want to critical check there closing comment. It looks that fibres from the material characteristics are comparable to plastic. -1- It would be good to say which type of fibres are considered. -2- if fibres do not emit CO2 is plastic emitting CO2? And when degrading fibres are emiting CO2, plastic not because it is not degrading. Are the listed mechanical attributes, comparable good as the one of plastic? (e.g., if they are biodegradable, the might be less stable).

Reviewer 2 Report

Please see attached PDF.

Overall, the quality of English language is good. There are some grammatical errors that should be corrected, for example:    

·       Line 146, “…illegally incineration plastics on-site…”: It should be, “…illegal incineration of plastics on-site…”

·       Line 486, “…, which making the new product…”: It should be, “which makes the new product…”

·      Use of "After”, like in "After OECD..." (Line 389). The more appropriate preposition is “According to OECD...” 

Another comment is the inconsistency in the format where some references are cited in the text and some are not. The pattern that I observed is that when a reference is mentioned at the beginning of the sentence, then it will be cited within the text; for example, "Gao, Yan et al [24] examined..." (Line 47); or, "In Germany, Bertling et al [38] showed..." (Line 75). However, in Line 234, the sentence begins with “According to [99], …” Based on the sentence structures in previous pages of the manuscript, the reference of [99] should be cited within the text. For consistency, please follow this format in the rest of the manuscript.

I would suggest that the paper be reviewed by a copyeditor or another colleague, focusing on grammar, punctuation, and sentence structure or format. Most of the time, a fresh pair of eyes is very helpful in catching errors.
